# The Antibacterial and Wound Healing Properties of Natural Products: A Review on Plant Species with Therapeutic Potential against *Staphylococcus aureus* Wound Infections

**DOI:** 10.3390/plants12112147

**Published:** 2023-05-29

**Authors:** Ana Elisa Belotto Morguette, Guilherme Bartolomeu-Gonçalves, Gabriella Maria Andriani, Giovana Elika Silveira Bertoncini, Isabela Madeira de Castro, Laís Fernanda de Almeida Spoladori, Ariane Mayumi Saito Bertão, Eliandro Reis Tavares, Lucy Megumi Yamauchi, Sueli Fumie Yamada-Ogatta

**Affiliations:** 1Programa de Pós-Graduação em Microbiologia, Departamento de Microbiologia, Centro de Ciências Biológicas, Universidade Estadual de Londrina, Londrina 86057-970, PR, Brazil; ae.belotto@gmail.com (A.E.B.M.); mgabriella.andriani@gmail.com (G.M.A.); isabela.mcastro@uel.br (I.M.d.C.); lais.spoladori@gmail.com (L.F.d.A.S.); lionilmy@uel.br (L.M.Y.); 2Programa de Pós-Graduação em Fisiopatologia Clínica e Laboratorial, Universidade Estadual de Londrina, Londrina 86038-350, PR, Brazil; guilherme.bartolomeu@uel.br; 3Laboratório de Biologia Molecular de Microrganismos, Universidade Estadual de Londrina, Londrina 86057-970, PR, Brazil; gih.bert15@gmail.com (G.E.S.B.); tavares.eliandro@uel.br (E.R.T.); 4Centro Universitário Filadélfia, Londrina 86010-520, PR, Brazil; ariane.bertao@unifil.br

**Keywords:** antibacterial activity, herbal products, methicillin-resistant *Staphylococcus aureus*, wound healing, wound infection

## Abstract

Wounds of an acute or chronic etiology affect millions of people worldwide, with increasing prevalence every year. Microbial infections are one of the main causes that impair the wound healing process, and *Staphylococcus aureus*, a commensal member of the skin microbiota, is one of the main causative agents of wound infections. Crucially, a high proportion of these infections are caused by methicillin-resistant *Staphylococcus aureus*, which, in addition to β-lactams, has acquired resistance to almost all the antibacterial agents used to treat it, limiting therapeutic options. Studies on the antimicrobial and healing activities of extracts, essential oils, or metabolites obtained from native plants have been reported in many countries that have a diverse flora and traditions with the use of medicinal plants for the treatment of wound infections. Due to their great chemical diversity, plants have proven to be promising sources of bioactive molecules for the discovery and development of new drugs or strategies for the treatment of wounds. This review highlights the main herbal preparations that have antimicrobial and healing activities with potential for the treatment of wound infections caused by *Staphylococcus aureus*.

## 1. Introduction

The skin is the organ most exposed to the environment and acts as the first line of defense against microbial pathogens. This complex and dynamic organ is inhabited by a wide range of commensal microorganisms, which can potentially shift to a pathogenic lifestyle, owing to both microorganism-related factors, such as the expression of virulence factors, and host-related factors (genetic factors, metabolic rate, age, and a loss of skin barrier integrity) [1,2]. The loss of skin integrity due to the presence of acute or chronic wounds can expose the subcutaneous tissues to colonization and infection by microorganisms, and this illness affects millions of people globally [3,4].

The etiology of wounds is quite varied and the evolution of their healing depends on several factors, such as good local vascularization and those related to patient conditions: age, obesity, diabetes, nutritional and immunological status, and the use of medications [3]. Acute wounds are caused by external damage, such as surgical cuts, burns, abrasions, or lacerations, and in general, they can heal naturally within 14 to 30 days [3,5]. However, persistent skin wounds due to individual conditions, contamination by microorganisms, or repetitive trauma can lead to a chronic condition that is difficult to heal [3,6,7].

For the management of wounds, strategies aimed at the prevention and treatment of infections, as well as the promotion of healing, should be employed [8] to strike a balance between inflammation and tissue formation and remodeling. Most infections can be prevented with the proper employment of antimicrobial prophylaxis. Nevertheless, an unintentional adverse effect caused by this is the selection of antimicrobial-resistant microorganisms, which accounts for a significant risk of developing chronic wounds due to persistent and difficult-to-treat infections. Furthermore, most antimicrobials used in clinical practice nowadays do not present antibiofilm activity [6,9,10,11].

Topical antiseptics also have limitations, since the most used antiseptics can harm the host cells, interfering with wound healing [8]. Thus, studies on the development of new strategies for the management of wounds are essential, with a particular focus on the prevention of microbial colonization and biofilm formation, as well as the ability to stimulate healing [12].

Among the bacterial components of the human microbiota, *Staphylococcus aureus* contribute to the greatest burden of skin infections in both surgical sites and chronic wounds [6,12,13,14], representing a significant public health problem, mainly attributed to the emergence of methicillin-resistant isolates [15,16]. In fact, this bacterial species can permanently or transiently colonize the skin of about 20–30% of the human population [17]. Critically, in addition to methicillin resistance, *S. aureus,* resistant to almost all available antimicrobials, has been identified in both colonization [18] and infection [19]. Moreover, among the several virulence and host immune evasion attributes of *S. aureus* [20], biofilm formation is an important factor that hinders wound treatment, impairing the inflammatory response, tissue regeneration, and epithelialization [12,21]. The biofilm consists of communities of cells tightly adhered to a surface and embedded within an exopolymeric matrix produced by these cells. Sessile cells within these communities exhibit an altered phenotype regarding their growth rate and gene transcription compared to free-floating planktonic cells [10]. Clinically important, biofilms exhibit a reduced sensitivity to antimicrobial agents (including topical antiseptics) and host defense mechanisms [9,10,20].

Natural products, which have been used for centuries in popular medicine due to their anti-inflammatory and healing properties [22,23,24], have renewed the interest of researchers because they present a vast diversity of bioactive molecules in their compositions. These compounds can be incorporated into pharmaceutical formulations containing natural or synthetic polymers compatible with host cells to develop new strategies for combating wound infections [3,24]. Using natural compounds to develop new drugs for wounds may contribute to a decrease in antimicrobial use, hospitalization, and costs related to healthcare assistance, improving the life quality of the affected patients.

## 2. *Staphylococcus aureus*

The genus *Staphylococcus* (Greek staphyle, “a bunch of grapes”; kokkos, “grain or berry”) belongs to the family Staphylococcaceae and is composed of catalase-producing Gram-positive cocci with diameters ranging from 0.5 to 1.5 µm. The term *Staphylococcus* was first proposed by Sir Alexander Ogston, who associated this bacterium with a suppurative abscess in a knee joint in the 1880s [25]. They are non-motile and non-spore-forming bacteria that can occur singly, in pairs, in tetrads, or in short chains that form grape-like irregular clusters [26]. Historically, species of this genus are classified as coagulase-positive or coagulase-negative *Staphylococcus*, according to their ability to produce coagulases under laboratory conditions [25].

Among the coagulase-positive staphylococci, *S. aureus* is the most prevalent in human infections worldwide [13,14]. This bacterium is part of the normal human microbiota, colonizing many sites in the body, but the anterior nostrils are its most frequently colonized niche [17]. This species is a facultative non-fastidious anaerobe bacterium that can grow at temperatures ranging from 15 to 45 °C and in NaCl concentrations of up to 15%. Due to the production of the yellow pigment staphyloxanthin, *S. aureus* presents typical golden-colored colonies on rich media, such as tryptic soy agar, and a hemolytic halo when cultivated on blood (from sheep or rabbit) agar. The mannitol salt agar containing 7.5% NaCl has been used as a selective and differential medium, since this microorganism can grow in high concentrations of salt and ferment mannitol, resulting in acid production [27].

An important feature of *S. aureus* is its ability to acquire resistance to almost all antimicrobials. In this sense, resistance to penicillin [28] and methicillin [29] emerged shortly after the introduction of these beta-lactam agents into clinical practice. The antimicrobial resistance of *S. aureus* has been linked to the acquisition of mobile genetic elements (MGE) carrying resistance-encoding genes and mutations affecting the key metabolic processes. These events can lead to various resistance mechanisms, including enzymatic drug inactivation, target modification, the active protection of the target site, and the active efflux of the antimicrobial agent [30,31,32].

The staphylococcal cassette chromosome *mec* (SCC*mec*), an MGE carrying the *mec* (*A* or *C*) and site-specific cassette chromosome recombinase (*ccr*) genes (*ccrAB* or/and *ccrC*), plays a pivotal role in the antimicrobial resistance of *S. aureus* [32]. The expression of a penicillin-binding protein (named PBP2a or PBP2′) with a lower affinity for β-lactam antibiotics, encoded by the *mec* genes, is the main mechanism of β-lactam resistance from methicillin-resistant *S. aureus*—MRSA [32]. Since the 60s, MRSA has become the leading cause of bacterial infections in healthcare settings and the community, spreading globally [15]. From the first report and up to the 90s, infections caused by MRSA used to be related to healthcare services (HA-MRSA, healthcare-associated-MRSA). However, from this decade onwards, community-acquired MRSA (CA-MRSA) infections in individuals with no previous hospitalization have been reported worldwide [33]. Remarkably, a high proportion of MRSA has acquired resistance to other non-β-lactam agents, including the last resort antimicrobials (glycopeptides, oxazolidinones, and daptomycin), which has crucial implications for the treatment options for this pathogen. Recent reviews on the molecular mechanisms of resistance to non-β-lactam agents are available in the literature [30,31].

Besides the antimicrobial resistance of planktonic cells, the production of biofilms by *S. aureus* on biotic (human epithelial mucosa) and abiotic (medical implanted devices) surfaces leads to persistent and difficult-to-treat infections [9,20]. In particular, this mode of microbial growth has been considered as one of the factors responsible for the chronicity of wounds [12]. In general, biofilm formation occurs through a complex process involving three main steps: (a) adhesion to biotic or abiotic surfaces; (b) the multiplication, accumulation, and maturation of the biofilm architecture; and (c) the detachment and dispersal of bacterial cells to a new surface [10,20]. In the initial stage of biofilm formation, which corresponds to adhesion, the bacteria can adhere to the surface, aided by the nonspecific factors (hydrophobicity, van der Waals forces, and electrostatic charges), environmental factors (temperature, pH, exposure time, cell density, chemical treatment, or the presence of antimicrobials), and physicochemical characteristics of the bacteria, such as bacterial surface proteins (adhesins) and host extracellular matrix proteins (collagen, fibronectin, fibrinogen, and vibronectin) [10]. Adhesins are encoded by several genes, many of which belong to the microbial surface components recognizing the adhesive matrix molecules (MSCRAMM) family [34]. In the second stage, the multiplication and accumulation of bacteria occur concomitantly with the production of an extracellular polymeric matrix (EPM). In *S. aureus*, this stage involves the production of the polysaccharide intercellular adhesin (PIA), also known as poly-*N*-acetylglucosamine (PNAG), according to its chemical composition, which promotes intercellular aggregation during the maturation of PIA-dependent biofilm formation [35]. In addition, extracellular DNA (eDNA) and other adhesive matrix components (proteins) form the biofilm’s EPM [10]. Finally, the structuring of biofilms is mediated by the activity of surfactant-like phenol soluble modulins (PSM) molecules, which disrupt the non-covalent interactions of the EPM. The three-dimensional structure of mature biofilms with channels is essential for delivering nutrients to the deeper layers of the community. Furthermore, PSMs, nucleases, and proteases play important roles in detaching cell clusters from biofilms to disperse/spread them to distal surfaces [10,20].

Given this scenario, the World Health Organization classified MRSA and vancomycin-resistant *S. aureus* (VRSA) as high-priority pathogens for developing new therapeutic strategies, highlighting the urgency concerning research into new antimicrobial agents capable of eradicating both planktonic and sessile cells [36].

## 3. Products of Natural Origin as Source of New Antimicrobials

Wound treatment includes clinical and/or surgical methods that aim to assist in the healing process in the shortest time [9,37]. In general, primary care is based on cleaning and debridement to remove foreign bodies and devitalized tissue; reducing odor, pain, and inflammation; preventing microbial colonization and infection; and inducing epithelialization and granular tissue formation. Dressing is one of the most used strategies for the treatment of wounds and aims at favoring the wound healing process, as well as protecting against new external aggressions [38]. Data from the literature show that dressings incorporated with natural products, such as honey [39], and those obtained from plants [24] have the potential for treating wounds. One of the advantages of using natural products in dressings is their chemical diversity, enabling a synergistic, pharmacological effect, especially in relation to anti-inflammatory, antioxidant, antimicrobial, and healing activities [24].

Plants have resisted their natural enemies for millions of years, with innate immune defense mechanisms that have evolved over time. Concerning microbial pathogens, plants have evolved mechanisms linked to pathogen recognition and defense responses [40]. In general, plants produce enzymes and secondary metabolites, reinforce their cellular structures in response to unfavorable environments, and defend themselves against predators [41]. Understanding how these species use their rich chemical compositions as a defense would reveal various compounds with different pharmacological properties [42]. In fact, due to their great diversity and recurrent use in popular medicine, natural products are targets of interest for researchers and the pharmaceutical industry, and are promising sources for isolating new antimicrobial molecules. Several studies have described the antibacterial activity of natural products on *S. aureus*, providing alternatives for the development of new drugs and products and, in this sense, combating microbial resistance. The activity of these compounds and their mechanism of action still need to be investigated in different microorganism species, including *S. aureus* [24,42,43]. However, the most significant challenges in these studies involve the search for a natural product with a low minimum inhibitory concentration (MIC), low toxicity to mammalian cells, and high bioavailability, ensuring the safety of its use [22,23,24].

Several products obtained from plants have been tested against *S. aureus* isolated from wounds, with the capacity to inhibit the growth of planktonic cells and biofilms in vitro and in wound infection models. Plant-derived natural products are characterized by a wide diversity of chemical classes and structural complexity. Terpenes, anthraquinones, tannins, flavonoids, lipids, and polysaccharides are some examples of chemical classes whose antibacterial activity has been reported in different species of bacteria [44,45]. Several physicochemical characteristics can influence the mechanism of the antibacterial activity of these compounds, including molecular mass, solubility, polar surface area, and the presence of different substituents [45]. Overall, these compounds can interfere with multiple bacterial targets, including cell wall synthesis, cell membrane integrity, DNA replication, protein synthesis, or enzyme activity [44]. Most of the studies presented in this review have reported the main phytochemicals of plant extracts, which generally belong to these chemical classes.

Table 1 shows the plants that have already been studied for their antibacterial potential in *S. aureus* isolated from wounds (in vitro) and/or in animal models of wounds infected with *S. aureus* (in vivo assay), covering the period from January 2015 to March 2023. The results of the studies presented in Table 1, which used traditional pharmaceutical formulations incorporated with natural plant products and evaluated their antibacterial effects in in vivo assays, are summarized in topic 4 of this review. Studies that only performed in vitro antibacterial activity assays against *S. aureus* and/or did not evaluate the pharmaceutical formulations incorporated with natural plant products in in vivo assays are summarized in Appendix A.

Current wound dressings include foams, films, hydrocolloids, ointment, cream, and hydrogels developed with biocompatible materials that allow for the incorporation of bioactive compounds with different pharmacological properties, which can be released directly at the wound site [24]. A wide variety of plants, native to different regions of the world, are being investigated for their potential as sources of bioactive compounds for the treatment of wound infections. Thus, new potential pharmaceutical formulations incorporated with natural products that have been tested against *S. aureus* and/or in experimentally induced wounds non-infected or infected by *S. aureus* are also presented in this review, and the results of the selected studies are described in the last topic (5).

## 4. Plant Extract-Based Dressings with Potential to Treat *Staphylococcus aureus* Infected Wounds

Ezzat et al. [56] described the therapeutic effect of an ointment containing 5.0% or 10.0% of an ethanolic extract or *N*-hexane or ethyl-acetate fractions of *Bergia ammannioides* Henye ex Roth. (Elatinaceae family) on excision wounds in Sprague Dawley male rats and Swiss albino mice. Treatment with the ointments (once a day/10 days) caused dose-dependent healing activity observed with all the ointments, and wound contraction percentages of 71.77, 85.62, and 81.29% were observed for the ethanolic extract, *N*-hexane, and ethyl-acetate fractions, respectively. In addition, an increase in collagen content was observed at the wound site. These results were comparable with those observed for Dermazine^®^ cream (reference drug). Moreover, the plant extract and fractions presented anti-inflammatory activity, inhibiting the topical acute edema induced by xylene in the mouse ears.

Su et al. [65] evaluated the therapeutic effect of an ointment (vaseline as a vehicle) containing a hydroethanolic extract (0.5 mg) from the leaves of *Entada phaseoloides* (L.) Merr. (Leguminosae family) on excision wounds infected with *S. aureus* ATCC 25923 in SD male rats. The application of the ointment (once a day/21 days) resulted in complete wound healing after 14 days of treatment and this effect was comparable to that observed for Bactroban (reference drug) treatment. A histological analysis of the wound tissue showed a regular epithelium organization enriched with fibroblast cells and collagen fibers. This effect may be due to the antibacterial and proliferative activity of the tannins present in the plant extract.

Karunanidhi et al. [48] evaluated the healing and antibacterial effects of ointments containing hexane or dichloromethane extracts (1.0%, 2.0%, or 5.0% *w*/*w*) obtained from the fresh bulbs of *Allium stipitatum* (Amaryllidaceae family) on burn wounds infected with MRSA ATCC 43300 in BALB/c female mice. The topical treatment with 20 mg of each ointment was carried out twice a day/20 days, starting 24 h post-infection. The ointments at 2.0% and 5.0% of both extracts caused 100% wound contraction, and a histological analyses revealed complete dermal epithelialization, intense fibroblast proliferation, and an absence of edema and necrosis after 15 days of treatment. Moreover, the count of MRSA colony-forming units (CFU) was zero after 72 h in the animal groups treated with both extracts.

An improved healing process was also observed in wounds non-infected or infected with MRSA ATCC BAA-1556, which were induced using the excision method in hairless female mice (Crl: SKH1-*Hrhr*) after treatment with a 10-major lipid mixture of lipid extract from *Chamaecyparis obtuse* (Cupressaceae family) [61]. The treatment of these wounds consisted of the topical application of a 0.02% lipid mixture three times a day/10 days. The lipid mixture accelerated wound healing; decreased the expression of pro-inflammatory mediators (TNF-α, IL-α, and IL-6) and the bacterial load; and increased the expression of IL-10 and olfactory receptor 520 (murine ortholog of OR2AT4), which affects the production of antimicrobial peptides, contributing to *S. aureus* killing [91], as well as keratinocytes migration, proliferation, and regeneration [92] at the wound site.

The study of Muller et al. [86] reported the application of a carbomer-based gel (Carbopol^®^) containing a 0.2% or 2.0% methanolic extract obtained from the leaves of *Sebastiania hispida* (Mart.) Pax (Euphorbiaceae family) on excision wounds infected with MRSA in Wistar rats. The topical application of the gels (once a day/3 or 21 days) was compared with Kollagenase^®^ ointment (reference drug). Both gels presented similar wound healing activity after 21 days and a histological analysis of the granulation tissue showed an organized and regenerated epithelium with increased collagen content. These effects may be attributed to the phenolics, flavonoids, and triterpenes compounds present in plant extracts.

The study of Martinez-Elizalde et al. [57] evaluated the therapeutic efficacy of a carbomer-based hydrogel (Carbopol^®^ 940) containing 10.0% of a methanolic extract from the barks of *Cyrtocarpa procera* Kunth (Anacardiaceae family) and *Bursera morelensis* (Burseraceae family) on incisional wounds in CD-1 male mice. The treatment consisted of a 0.5 mL gel application twice a day/10 days. Both plant extracts presented healing activity similar to the reference drug Recoveran, presenting increased neovascularization and regenerated epithelia.

Tatiya-Aphiradee et al. [66] described the healing and antibacterial activity of an ethanolic extract obtained from the fruit pericarp of *Garcinia mangostana* Linn (Clusiaceae family) on a tap-stripping model (skin superficial infection) infected with MRSA DMST 20651 in ICR male mice. The topical formulations (100 µL) were prepared in 10.0% ethanol in propylene glycol (vehicle), containing 10.0% ethanolic extract or 1.32% α-mangostin (major xanthone in the fruit pericarp), and were applied to the wounds 24 h after bacterial infection once a day/10 days. The treatment with the ethanolic extract and erythromycin (reference drug) caused complete wound healing at day 10, and a significant reduction in the bacterial load was also observed. The histological analyses showed almost a complete regeneration of the epithelium at day 8 and no inflammatory cell infiltration. A decreased mRNA expression of the pro-inflammatory cytokines (TNF-α, IL-6, and IL-1β) and Toll-like receptor 2 (TLR-2), which is involved in *S. aureus* recognition [93], was also observed. The treatment with α-mangostin was less efficient than both formulations and did not down-regulate the expression of the pro-inflammatory cytokine genes.

The efficacy of a topical application (once a day/16 days) of ointments containing 2.0% or 4.0% of *Anethum graveolans* (Apiaceae family) essential oil was evaluated on excisional wounds infected with MRSA in BALB/c female mice. The ointments increased the expression of p53, which induces the apoptosis of inflammatory cells and then reduces the inflammatory phase of wound healing. Moreover, the ointment treatment increased both the release of caspase-3 from apoptotic cells and the expressions of Bcl-2 (an integral outer mitochondrial membrane protein), vascular endothelial growth factor (VEGF), and fibroblast growth factor (FGF-2), which induces accelerated epithelialization, angiogenesis, and fibroblast and collagen deposition. These effects were more pronounced when compared to those caused by the mupirocin (reference drug) treatment. After 16 days of the ointment treatment, 100% wound contraction was observed and no MRSA CFU was detected [51].

Güzel et al. [83] evaluated the effect of ointments containing 0.5% or 1.0% of ethanolic extracts from *Salvia kronenburgii* Rech. f. and *Salvia euphratica* Montbret, Aucher & Rech. f. var. *euphratica* (Lamiaceae family) on excision and incision wounds in Wistar diabetic male rats. The ointment application was performed once a day/7 or 14 days. Wound contraction percentages of 99.9%, 99.5%, and 99.7% were observed for 0.5% and 1.0% of the ethanolic extract from *S. kronenburgii* (SK) and 1.0% of that from *S. euphratica* (SE), respectively. For the incision model, 99.5% and 99.7% contraction occurred after the treatment with the SK and SE ointments, respectively. Increased epithelialization, angiogenesis, and collagen content and reduced inflammation were observed at the wound site for both plant extracts. These results were comparable with the Fito^®^cream containing a 15% (*w*/*w*) *Triticum vulgare* L. aqueous extract (reference drug) treatment. The SE ointments (0.5% and 1.0%) on day 7 and SK ointment (0.5%) on day 14 reduced the oxidative damage to DNA.

The study of Farahpour et al. [84] reported the therapeutic efficacy of an ointment containing essential oil (2.0 or 4.0%) from *Salvia officinalis* L. (Lamiaceae family) in excision wounds infected with *S. aureus* ATCC 25923 in BALB/c mice. The treatment was performed once a day/14 days, starting after the infection with *S. aureus*. The ointment containing 4.0% of the essential oil showed better results related to the healing process, including when compared to those observed with the mupirocin (reference drug) treatment, such as a reduced wound area, bacterial load, and pro-inflammatory cytokine mRNAs (IL-1β, IL-6, and TNF-α), and an increased fibroblast migration, collagen deposition, angiogenesis, and tissue total antioxidant capacity, which is essential to preventing the establishment of chronic wounds [94].

Yang et al. [52] reported the healing efficacy of a mixture (1:1) of ethanolic extracts from *Angelica dahurica* Bentham et Hooker (Umbelliferae family) and *Rheum officinale* Baill. (Polygonaceae family) on excision wounds infected with *S. aureus* ATCC 29213 in Sprague Dawley male rats. The topical treatment (0.2 mL) with the extract (once a day/7 days) accelerated the wound healing, which was visualized by an increased wound contraction, epithelialization, and angiogenesis, and a reduction in the bacterial load and plasmatic levels of TNF-α and IL-6.

The antibacterial and healing activities of an ethanolic extract from the leaves of *Piper betle* L (Piperaceae family) were evaluated on excision wounds infected with *S. aureus* ATCC 25923 in BALB/c male mice [79]. The treatment was carried out once a day/14 days with an aqueous cream containing 2.5% or 5.0% of plant extract, and mupirocin cream was used as a reference drug. A significant percentage of wound contraction was observed for both plant extract concentrations compared to the untreated control (around 90.0% for plant-extract-treated wound versus 40.0% for untreated wound). In addition, intense epithelialization, fibroblast migration, and collagen deposition were observed in the dermal layer of the plant-extract-treated wound. These results were comparable with those presented with the mupirocin treatment. Moreover, the count of *S. aureus* CFU was zero in the animal groups treated with the cream containing 5.0% of *P. betle* extract or mupirocin.

Ekom et al. [59] developed a carbomer-based gel containing 1.0%, 5.0%, or 10.0% of methanolic extract from the dried fruits of *Capsicum annuum* L. (Solanaceae family). The therapeutic effect of the gels was analyzed in excision wounds infected with *S. aureus* in Wistar albino rats. The treatment consisted of topical application to the wounds and started 24 h after the establishment of infection. After 20 days of treatment, there was complete healing of the wounds and the count of *S. aureus* CFU was zero in the animal groups treated with the hydrogels containing 5.0% and 10.0% of *C. annuum* extract, as occurred with the clindamycin (Aclin gel, reference drug) treatment. Similar results were obtained by the same researchers using a gel containing a methanolic extract (5.0 or 10.0%) from the seeds of *Persea americana* Mill. (Lauraceae family) in excision wounds infected with *S. aureus* in Wistar albino rats [78].

The study of Rajoo et al. [64] evaluated the therapeutic effect of an ointment containing 10.0% of a methanolic extract from the leaves of *Elaeis guineensis* Jacq. (Arecaceae family) on wounds infected with *S. aureus* in Sprague Dawley male rats. BETADINE^®^, a commercial ointment containing povidone-iodine (10.0% *w*/*v*), was used as a reference drug. The treatment (once a day) was performed over 20 days, starting 24 h after the establishment of an *S. aureus* infection. Compared to the *S. aureus* infected and untreated wound animal groups, the treatment with the ointment plus the *E. guineensis* extract and BETADINE^®^ enhanced the migration of fibroblasts and epithelial cells to the wound site, increasing the collagen fiber content. A significant reduction in the bacterial load at the wound site was also observed.

Improved healing activity was also observed for a biocompatible ointment containing a methanolic extract from the leaves from *Moringa oleifera* Lam (Moringaceae family) in excision wounds infected with MRSA ATCC 43300 in non-diabetic [73] and diabetic [74] Wistar rats. The formulation containing 20.0% of the methanolic extract (*w*/*w*) increased angiogenesis, collagen deposition, and the activity of the antioxidant enzymes, superoxide dismutase and catalase, and reduced the bacterial load and epithelialization period at the wound site. However, the mupirocin (reference drug) treatment showed better results related to the healing process when compared to the ointment containing the methanolic extract.

An accelerated contraction of wounds and reduction in the bacterial load at the wound site were also observed after a treatment with an ointment containing clove oil (5.0% or 10.0% *w*/*w*) from *Syzygium aromaticum* (Myrtaceae family). The ointments were applied once a day on excision wounds infected with MRSA in Wistar albino rats, starting 24 h after the bacterial inoculation. Imipenem (reference drug) and clove oil (10% *w*/*w*) reduced the bacterial load and improved the epidermal regeneration and angiogenesis at the wound site. Although the combination of 10.0% clove oil and imipenem also improved the epithelial regeneration of the wounds, there was no significant difference when compared to the imipenem or clove oil monotherapy, indicating a lack of interaction between both compounds [87].

## 5. Emerging New Herbal-Based Dressings with Potential to Threat *Staphylococcus aureus*-Infected Wound

In addition to classic pharmaceutical formulations, several researchers have developed new materials containing products extracted from plants with antimicrobial and healing properties, with the potential for treating infected wounds. In this topic, we will present the selected materials, in which at least one natural product extracted from plants was used in their preparation and an evaluation of their antibacterial activity against *S. aureus* and/or application to wounds non-infected or infected with this bacterial species (Table 2) was performed. In general, the in vitro antibacterial activity of the new materials was evaluated by diffusion-based assays, except when specified.

Tummalapalli et al. [95] developed a biocomposite membrane consisting of a nonwoven cotton fabric loaded with oxidized pectin-gelatin-*Aloe vera* (Asphodelaceae family) leaf extract and/or curcumin. The biodegradable biocomposite membranes loaded with *A. vera* (40.0%) or curcumin (40.0%) presented antibacterial activity against *S. aureus*, reducing the bacterial CFU count. The *A. vera* membrane exhibited a higher biocompatibility to the mouse NIH3T3 cells than the curcumin membrane. The combination of *A. vera* and curcumin did not improve the antibacterial and cytotoxicity activities. In addition, wound contraction percentages of 80%, 60%, and 40% were observed for the treatment with the *A. vera*, curcumin, and *A. vera plus* curcumin membranes, respectively, in an excision wound model in C57BL/6J mice. Organized collagen deposition and neovascularization were also observed in the *A. vera*-treated wounds.

Hybrid nanofibers of poly-L-lactic acid and gelatin loaded with *Lawsonia inermis* (Lythraceae family) were prepared using the electrospinning method. The nanofibers presented no cytotoxicity to NIH3T3-L1 fibroblast cells and a gradual release of *L. inermis* from the material was also observed. A significant reduction in *S. aureus* ATCC 25923 CFU counts was detected after 2 h of incubation in the presence of the nanofibers [96].

The study of Singh et al. [97] developed a biocomposite membrane consisting of a dextran-nanosoy-glycerol-chitosan base matrix loaded with clove oil or sandalwood oil. The clove oil (5.0%)-loaded membrane and sandalwood (10.0%)-loaded membrane presented antibacterial activity against *S. aureus*, reducing the bacterial CFU count. In addition, a reduction in the bacterial adhesion to these membranes was observed. Both herbal-loaded membranes accelerated the healing process in the excision wound model in the BALB/c mice, promoting fibroblast migration and collagen deposition at the wound site.

Fayemi et al. [72] developed polyacrylonitrile (16.0% wt)-based nanofibers incorporated with leaf extracts from *Moringa oleifera*. The addition of a plant extract (0.10, 0.15, 0.25, or 0.5 g) caused concentration-dependent antibacterial activity against *S. aureus* and *Escherichia coli*. Moreover, the healing activity was analyzed on excision wounds in Wistar rats. The nanofibers with 0.5 g of plant extract (replaced every two days) improved the healing process, with a complete re-epithelialization and organized arrangement of the collagenous fibers of the wounded skin observed on day 11.

Ardekani et al. [90] developed polyvinyl alcohol (PVA)-based nanofibers loaded with essential oil from *Zataria multiflora* (Lamiaceae family) using the electrospinning method. The essential oil (2.0, 5.0, or 10.0%) was loaded into a 6.0% PVA/3.0% chitosan/gelatin solution and then electrospun into uniform nanofibers. This material exhibited a swelling capacity and no cytotoxicity for L929 murine skin fibroblast cells. Moreover, the nanofibers presented antimicrobial activity, reducing the count of CFU to zero, in the preparations containing 2.0, 5.0, and 10.0% of the plant extract, of *Candida albicans* ATCC 10261, and 5.0 and 10.0% for *S. aureus* ATCC 25923 and *Pseudomonas aeruginosa* ATCC 27853.

By using the electrospinning method, Ali et al. [53] developed a dual-layer nanofibrous membrane composed of: (a) 3.0% chitosan loaded with a methanolic extract from the leaves of *Azadirachta indica* (Meliaceae family) as a coating layer; and (b) 10.0% PVA as an inner layer. The membrane exhibited porosity, wettability, an enhanced thermal stability, and antibacterial activity against *S. aureus*, which are essential in wound healing [98].

The study of Khezri et al. [82] described the therapeutic effect of the essential oil of *Rosmarinus officinalis* L. (Lamiaceae family) encapsulated into nanostructured lipid (glyceryl palmitostearate as a solid lipid and miglyol as a liquid lipid) carriers. The in vitro antibacterial activity of the functionalized lipid nanocarriers was evaluated against several reference (ATCC) bacterial strains, including *S. aureus* ATCC 25923, and the growth inhibition was more pronounced in the Gram-positive bacteria. Moreover, a carbomer-gel (Carbopol^®^ Ultrez 21) containing the nanoencapsulated essential oil (treatment once a day/14 days, starting 24 h post-infection) accelerated the healing of *S. aureus*-infected wounds induced in BALB/c male mice, reducing the bacterial load and increasing the re-epithelialization, angiogenesis, and fibroblast cell migration at the wound site. These results were comparable with those observed for the Mupirocin^®^ ointment, which was used as a reference drug.

Zepon et al. [99] developed a pH-responsive hydrogel film based on carrageenan, locust bean gum, and cranberry extract. The hydrogel exhibited moderate antibacterial activity against *S. aureus*. A reduction in the bacterial adhesion to this film was also observed. The film presented a dose-dependent biocompatibility with NIH3T3 cells.

Cellulose from the reproductive organs of *Gleditsia triacanthos* L. (Fabaceae family) was purified and functionalized with an ethanolic extract (enriched with phenolic compounds) obtained from the same plant. The functionalized cellulose microfibers presented wettability, biocompatibility with L929 cells, and antioxidant activity. In addition, this material inhibited the growth (determined by CFU counts) of one clinical MRSA isolate, Gram-negative bacteria (clinical isolates of *P. aeruginosa*, *Enterobacter cloacae*, *Acinetobacter baumannii*, and *E. coli* ATCC 11229), and fungal species (*C. albicans* ATCC 10231 and *Candida parapsilosis* ATCC 22019). A linear rate of the phenolic compound release from the cellulose microfibers was observed in the first 6 h, and an almost complete release of the phenolic compounds was detected within 120 h [65].

Mouro et al. [47] developed a biocompatible dual-layer nanofibrous membrane composed of: (a) cotton fabric oxidized with a 2,2,6,6-tetramethylpiperidine-1-oxyl radical/sodium bromide/sodium hypochlorite as substrate; and (b) PVA and chitosan-containing an ethanolic crude extract from *Agrimonia eupatoria* L. (Rosaceae family) as a coating layer. The composite material exhibited wettability, porosity, and swelling capacity. Moreover, the addition of 5.0% (wt) of the crude extract from *A. eupatoria* enhanced the bactericidal activity of this material against *S. aureus* and *P. aeruginosa* in vitro.

Epigallocatechin gallate (EGCG)-modified black phosphorus quantum dot (BPQD) nanoparticles were synthesized and evaluated as therapeutic nanoplatforms for healing burn wounds. The hydrogels containing EGCG-BPQD presented antibacterial activity against MRSA, reducing the bacterial CFU count and biofilm biomass. The antibacterial effect was enhanced by the near-infrared irradiation (phototherapy). This irradiation can increase the local temperature, promoting enzyme inactivation and disrupting the bacterial metabolism. In addition, EGCG-BPQD increased the intracellular ROS content, promoting cell membrane damage and protein leakage. Moreover, EGCG-BPQD induced fibroblast migration and proliferation and angiogenesis in a human umbilical vein endothelial cell (HUVEC) scratched monolayer, and was non-toxic to HUVECs. In a burn wound infected with MRSA in Sprague Dawley diabetic rats, the application of the hydrogel upregulated the CD31 (angiogenesis-associated protein) and basic fibroblast growth factor (bFGF) that can accelerate the wound contraction, in the absence of significant systemic damage to the animals [100].

Gerges et al. [101] developed an ointment containing 1.0% polygalacturonic and 0.4% caprylic acids with antibiofilm activity against MRSA. This effect was observed after 2 h of treatment, whereas benzalkonium chloride, MediHoney, and polyhexamethylene biguanide ointments failed to eradicate the biofilms. No toxicity for L929 fibroblast cells and bovine erythrocytes was observed after 24 h of incubation.

Gharibi et al. [102] synthesized a gelatin-based hydrogel loaded with a reactive methoxy-silane-functionalized quaternary ammonium compound bearing a fatty amide residue originating from castor oil (Si-CAQ). The biocompatible and hemocompatible hydrogel presented bactericidal activity against MRSA. Moreover, it showed wound healing activity in L929 fibroblast scratched cells and in vivo in Wistar rats by enhancing the epithelialization, collagen deposition, and vascularization at the wound site.

In the study of Rizg et al. [103], a geranium-oil-based self-nanoemulsifying drug delivery system loaded with pravastatin was developed. The nanoemulsion presented antibacterial activity against *S. aureus* in vitro. Moreover, the nanoemulsion exhibited anti-inflammatory effects and accelerated the wound contraction in a burn wound model in Wistar rats.

D’Souza et al. [55] developed a biocompatible film containing an aqueous extract obtained from fresh cut stems of *Basella alba* L. (Basellaceae family), using PVA and chitosan as a matrix. The film presented thermal stability, flexibility, wettability, swelling capacity, enhanced anti-inflammatory activity, compatibility with human blood, and no cytotoxicity to L929 cells in vitro. The addition of the plant extract (2.5, 3.5, or 4.5 mL) exhibited dose-dependent antibacterial activity against *S. aureus* and *E. coli* and healing activity in an L929 fibroblast scratched monolayer assay.

Finally, Li et al. [104] developed a bioactive hydrogel using the self-assembly behavior of saponin glycyrrhizic acid in aqueous solutions. The biocompatible and hemocompatible hydrogel presented in vitro and in vivo bactericidal activity against *S. aureus*. The hydrogel enhanced the formation of granulation tissue and collagen deposition and downregulated the inflammatory response, contributing to an improvement in the healing process in a full-thickness wound model in Kunming mice infected or non-infected with *S. aureus*.
plants-12-02147-t002_Table 2Table 2New dressings loaded with natural products with potential for the treatment of wounds infected by *Staphylococcus aureus*.MaterialPlant ProductFormulationBiological ActivityReferenceBiocomposite membranesLeaf extract of*Aloe vera*CurcuminOxidized pectin was crosslinked with gelatin to produce the matrix, which was mixed with *A. vera* (10.0–40.0% wt), curcumin, or aloe vera (20.0% and curcumin (20.0%).-In vitroAntibacterialNon-toxic NIH3T3 cells-Excision wound in C57BL/6J mouseWound contractionAngiogenesis[95]Electrospun nanofibers*Lawsonia inermis*(Lythraceae Family)Gelatin (20.0%) solution loaded with *L. inermis* (10.0 and 20.0% of weight ratio to gelatin) and 6.0% of poly-L-lactic acid solution were prepared and electrospun to produce the nanofibers.-In vitroAntibacterialNon-toxic NIH3T3-L1 cells[96]Biocomposite membranesClove oilSandalwood oilClove oil (2.0, 5.0, 10.0, or 15.0%) or sandalwood oil (2.0, 5.0, 10.0, or 15.0%) was loaded into dextran/nanosoy/glycerol/ chitosan base matrix.-In vitroAntibacterialInhibition of bacterial adhesion-Excision wound in BALB/c mouseRe-epithelialization process[97]Electrospun Nanofibers*Moringa oleifera* extract0.10, 0.15, 0.25, or 0.50 g of plant extract were dissolved into 16.0% (wt) polyacrylonitrile.-In vitroAntibacterial-Excision wound in Wistar ratWound contractionRe-epithelialization process[72]Electrospun NanofibersEssential oil of *Zataria multiflora*2.0, 5.0 or 10.0% of essential oil were mixed with a chitosan/poly(vinyl alcohol)/gelatin polymeric solution and then used to prepare the nanofibers.-In vitroAntibacterialNon-toxic to L929 cells[90]Electrospun dual-layer nanofibrous membraneMethanolic extract of *Azadiracta indica*Plant extract/chitosanblend was electrospun on poly(vinyl alcohol) nanofibrous form.-In vitroAntibacterial[53]Nanostructured lipid carrierEssential oil of *Rosmarinus officinalis*Essential oil was encapsulated into nanostructured lipid carrier (consisting of glyceryl palmitostearate as solid lipid and miglyol as liquid lipid) and loaded into a carbomer-based hydrogel.-In vitroAntibacterial-Excision wound in BALB/c male mouseAntibacterialRe-epithelialization processAngiogenesis[82]FilmKappa-carrageenanLocust bean gumCranberry extractA pH-responsive film based on kappa-carrageenan and locust bean gum polysaccharides was loaded with 0.3% cranberry extract.-In vitroAntibacterialInhibition of bacterial adhesionNon-toxic to NIH3T3 cells[99]Plant cellulose microfibersCellulose and phenolic compounds of *Gleditsia triacanthos*Cellulose microfibers were functionalized with phenolic compounds extracted from the leaves.-In vitroAntibacterialAntioxidantNon-toxic to L929 cells[67]Dual-layer nanofibrous membraneHydroethanolic extract of *Agrimonia eupatoria L*Poly(vinyl alcohol)/chitosan/plant extract (10.0% *w*/*v*) (second layer) was electrospun over a cotton fabric (first layer) membrane.In vitroAntibacterialNon-toxic to NHDF cells[47]Black phosphorus quantum dots (BPQD)Epigallocatechin gallate (EGCG)Epigallocatechin-gallate-modified BPQDs were loaded into hydrogels-In vitroAntibacterial, including antibiofilmNeovascularization and proliferation of HUVEC cells in scratch wound assayNon-toxic to HUVEC cells-Burn wound in Sprague Dawley diabetic ratWound contractionRe-epithelialization processNo systemic toxicity[100]OintmentCaprylic acidPolygalacturonic acid1.0% polygalacturonic acid plus 0.4% caprylic acid incorporated into 2-hydroxyethylcellulose and glycerol ointment base.-In vitroAntibiofilmNon toxic to L929 cells and bovine erythrocytes[101]HydrogelCastor oilA reactive methoxy-silane-functionalized quaternaryammonium compound bearing a long fatty amide residue originating fromcastor oil (Si-CAQ) was synthesized and then loaded into gelatin/poly(vinyl alcohol)-based hydrogel.-In vitroBactericidalNon toxic to L929 cellsProliferation of L929 cells in scratch wound assay-Excision wound in Wistar ratRe-epithelialization process[102]NanoemulsionGeranium oilGeranium oil was mixed with the surfactant (Tween80/Span80) to prepare the nanoemulsion. The pravastatin was loaded into the nanoemulsion.-In vitroAntibacterial-Burn wound in Wistar ratAnti-inflammatoryWound contraction[103]Composite film*Basella alba* aqueous extractPoly (vinyl alcohol)/chitosan/plant extract composite film.-In vitroAntibacterialAnti-inflammatoryProliferation of L929 cells in scratch wound assayNon-toxic to L929 cells[55]HydrogelGlycyrrhizic acid2.0% Aldehyde-contained glycyrrhizic acid (AGA) was mixed with 2.0% carboxymethyl chitosan.-In vitroBactericidalAnti-inflammatoryNon-toxic to L929 cells, RAW 264.7 macrophages and erythrocytes-Excision wound in Kunming mouseBactericidalAnti-inflammatoryRe-epithelialization process[104]NIH3T3 embryonic mouse fibroblast cells; L929: mouse skin fibroblast cells; NDHF: normal human dermal fibroblasts; and HUVEC: human umbilical vein endothelial cells.

## 6. Methods

The literature on plant natural products with potential antibacterial and wound healing activities in vitro and in vivo was searched in the science databases of PubMed, Scopus, and the Cochrane Library. Peer-reviewed papers covering the period from January 2015 to March 2023 were considered. Studies in which the origin of the tested *S. aureus* isolate could not be identified were excluded from the review. The search terms were “wound healing”, “plant”, “extract”, “natural products”, “antibacterial activity”, and “*Staphylococcus aureus*”. To contribute to the field, in this review, we focused on the plants and their derivatives that exhibit antibacterial activity in vitro against *S. aureus* (including MRSA), isolated from wound infections and/or in vivo models of wounds infected with this bacterium.

## 7. Conclusions and Future Directions

Several plant species have potential as a source of molecules capable of inhibiting the growth of *S. aureus* in vitro and in animal models of wound infection. In addition, they present healing properties in in vitro and in vivo models, exhibiting anti-inflammatory, antibacterial, antioxidant, and angiogenic activities, as well as stimulating collagen synthesis and cell proliferation. The active compounds can be found in extracts from different parts of the plant prepared with different solvents, in essential oils and oil resin, among others. Studies proving their effectiveness in treating wounds in humans are needed. The results may contribute to the development of new low-cost and accessible strategies for the treatment of wounds infected by *S. aureus*.

## Figures and Tables

**Table 1 plants-12-02147-t001:** Plant species with antibacterial and healing properties currently investigated for *Staphylococcus aureus* wound infections, covering from January 2015 to March 2023.

Species	Family	Plant Part	Biological Activity	Reference
*Achyranthus aspera* Telenge	Amaranthaceae	Leaves	-In vitro Antibacterial	[46]
*Agrimonia eupatoria* L.	Rosaceae	NI	-In vitro Antibacterial Non toxic to NDHF cells	[47]
*Allium stipitatum*	Amaryllidaceae	Fresh bulbs	-In vitro Antibacterial * Non toxic to Vero cells -Burn wound in BALB/c mouse Antibacterial Re-epithelialization process	[48]
*Aloe* spp. *A. tormentorii* (Marais) L.E.Newton and G.D.Rowley *A. purpurea* Lam. *A. macra* Haw. *A. lomatophylloides Balf.f* *A. vera*	Asphodelaceae	Leaves	-In vitro Antibacterial Proliferation of HaCaT cells in scratch wound assay Except *A. purpurea*, the other extracts were non toxic to HL60 and MCR 5 cells	[49]
*Althaea officinalis* L.	Malvaceae	Leaves	-In vitro Bactericidal -Excision wound in Wistar male rat Re-epithelialization process	[50]
*Anethum graveolens* L	Apiaceae	NI	-Excision wound in BALB/c mouse Antibacterial Anti-inflammatory Re-epithelialization process Angiogenesis	[51]
*Angelica dahurica* Benth. et Hooker.f *Rheum officinale* Baill.	Apiaceae Polygonaceae	NI	-In vitro Antibacterial -Excision wound in Sprague Dawley rat Antibacterial Anti-inflammatory Re-epithelialization process Angiogenesis	[52]
*Azadirachta indica*	Meliaceae	Leaves	-In vitro Antibacterial	[46,53]
*Balanites aegyptiaca*	Balanitaceae	Bark	-In vitro Antibacterial Antioxidant	[54]
*Basela alba* L.	Basellaceae	Fresh stem	-In vitro Antibacterial Proliferation of L929 cells in scratch wound assay Anti-inflammatory	[55]
*Bergia ammannioides* Henye ex Roth.	Elatinaceae	NI	-In vitro Antibacterial Antioxidant -Excision wound in Sprague Dawley rat and Swiss albino mouse Anti-inflammatory Re-epithelialization process	[56]
*Bursera morelensis*	Burseraceae	Barks	-In vitro Antibacterial Antioxidant -Excision wound in male CD-1 mouse Re-epithelialization process Angiogenesis	[57]
*Calophyllum inophyllum*	Calophyllaceae	Seeds	-In vitro Antibacterial Proliferation of HaCaT cells in scratch wound assay Induction of the antimicrobial peptide β-defensin 2 release by macrophages	[58]
*Capsicum annuum* L.	Solanaceae	Dried fruits	-In vitro Antibacterial -Excision wound in Wistar rat Antibacterial Non-toxic to skin and eyes Induction of wound contraction	[59]
*Carthamus tinctorius* L.	Asteraceae	Seeds	-In vitro Antibacterial Antioxidant	[60]
*Chamaecyparis obtuse*	Cupressaceae	NI	-In vitro Bactericidal -Excision wound in hairless female mouse (Crl: SKH1-Hrhr) Antibacterial Anti-inflammatory Re-epithelialization process	[61]
*Commiphora gileadensis*	Burseraceae	Leaves and branches	-Excision wound in BALB/c mouse Antibacterial Anti-inflammatory Re-epithelialization process	[62]
*Cratylia mollis*	Fabaceae	Seeds	-Infection using *Tenebrio monitor* larvae Increase infected-larvae survivor -Excision wound in Swiss mouse Antibacterial Anti-inflammatory Re-epithelialization process	[63]
*Cyrtocarpa procera* Kunth	Anacardiaceae	Barks	-In vitro Antibacterial Antioxidant -Excision wound in male CD-1 mouse Re-epithelialization process Angiogenesis	[57]
*Elaeis guineensis* Jacq.	Arecaceae	Leaves	-Excision wound in Sprague Dawley rat Antibacterial Re-epithelialization process	[64]
*Entada phaseoloides* (L.) Merr.	Leguminosae	NI	-In vitro Bactericidal Proliferation of NIH3T3 cells in scratch wound assay -Excision wound in male SD rat Re-epithelialization process	[65]
*Garcinia mangostana* Linn	Clusiaceae	Fruit pericarp	-In vitro Antibacterial -Tap stripping wound in ICR mouse Antibacterial Anti-inflammatory Re-epithelialization process	[66]
*Gleditsia triacanthos* L.	Fabaceae	Reproductive organs	-In vitro Antibacterial Antioxidant Non-toxic to L929 cells	[67]
*Hypericum perforatum*	Hypericaceae	Aerial parts	-In vitro Antibacterial	[68]
*Jatropha multifida* L.	Euphorbiaceae	Leaves	-In vitro Antibacterial Anti-inflammatory	[69]
*Jatropha neopauciflora* L.	Euphorbiaceae	Latex	-In vitro Antibacterial Antioxidant Non-toxic to Ca Ski ATCC CRL-1550 and NIH3T3 cells -Excision wound in CD1 male Mus musculus mouse Wound contraction -Carrageenan-induced edema in Wistar rat Anti-inflammatory	[70]
*Lawsonia inermis* Henna	Lythraceae	Leaves	-In vitro Antibacterial	[46]
*Moringa oleifera*	Moringaceae	Leaves	-Excision wound in Wistar rat Wound contraction	[71]
		Leaves	-In vitro Antibacterial -Excision wound in Wistar rat Re-epithelialization process	[72]
		Leaves	-In vitro Antibacterial -Excision wound in Wistar rat Antioxidant Re-epithelialization process Angiogenesis	[73,74]
*Nigella sativa* Linn	Ranunculaceae	Black seed	-In vitro Antibacterial	[75]
*Opuntia fícus-indica* Miller	Cactaceae	Flowers	-In vitro Antibacterial Antioxidant -Excision wound in Wistar male rat Re-epithelialization process Angiogenesis	[76]
*Parrotiopsis jacquemontiana*	Hamamelidaceae	Leaves	-In vitro Antibacterial Antioxidant -Excision wound in Sprague Dawley rat Re-epithelialization process	[77]
*Persea americana* Mill.	Lauraceae	Seeds	-In vitro Antibacterial, including antibiofilm -Excision wound in Wistar rat Antibacterial Wound contraction Non-irritant to skin and eyes	[78]
*Piper betle* L.	Piperaceae	Leaves	-In vitro Antibacterial -Excision wound in BALB/c mouse Antibacterial Re-epithelialization process	[79]
*Plukenetia volubilis* L.	Euphorbiaceae	Seeds	-In vitro Inhibition of bacterial adhesion Non-toxic to human keratinocytes nor human skin explant	[80]
*Portulaca oleracea*	Portulacaceae	Trunk and leaves	-In vitro Antibacterial -Excision wound in Kunming mouse Antibacterial Anti-inflammatory Wound contraction	[81]
*Quercus alba*	Fagaceae	Bark	-In vitro Antibacterial	[68]
*Rosmarinus officinalis* L.	Lamiaceae	Essential oil	-In vitro Antibacterial -Excision wound in BALB/c mouse Antibacterial Re-epithelialization process Angiogenesis	[82]
*Salvia euphratica* Montbret, Aucher and Rech. f. var. *euphratica* *Salvia kronenburgii* Rech. f.	Lamiaceae	Aerial parts	-In vitro Antibacterial Antioxidant -Excision and incision wound in male Wistar rat Antibacterial Anti-inflammatory Re-epithelialization process Angiogenesis	[83]
*Salvia officinalis* L.	Lamiaceae	Leaves	-In vitro Antibacterial Excision wound in BALB/c mouse Antibacterial Re-epithelialization process Angiogenesis	[84]
*Salvia sclarea* L.	Lamiaceae	NI	-In vitro Antibacterial	[85]
*Sebastiania hispida* (Mart.) Pax	Euphorbiaceae	Leaves	In vitro Antibacterial -Excision wound in Wistar rat Re-epithelialization process Angiogenesis	[86]
*Syzygium aromaticum*	Myrtaceae	Essential oil	-In vitro Antibacterial -Excision wound in Wistar rat Antibacterial Re-epithelialization process Angiogenesis Non-toxic to skin	[87]
*Urtica dioica*	Urticaceae	Leaves	-In vitro Antibacterial -Excision wound model in Wistar rat Re-epithelialization process Angiogenesis	[88]
*Zanthoxylum nitidum* (Roxb.) DC.	Rutaceae	Dried roots	-In vitro Antibacterial -Wound infection in Kunming mouse Antibacterial	[89]
*Zataria multiflora*	Lamiaceae	Essential oil	-In vitro Antibacterial	[90]

* Extract was non-toxic to mammalian cells at minimal inhibitory or bactericidal concentrations (MIC/MBC). NDHF: normal human dermal fibroblasts; HaCaT: human adult skin keratinocytes; HL60: human promyelocytic leukemia cells; MCR 5: human normal fibroblast cells; L929: mouse skin fibroblast cells; NIH3T3 embryonic mouse fibroblast cells; and Ca Ski: human cervical carcinoma cells.

## Data Availability

Not applicable.

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
