# Peer review of "The Antibacterial and Wound Healing Properties of Natural Products: A Review on Plant Species with Therapeutic Potential against Staphylococcus aureus Wound Infections"

_plants, 2023, doi:10.3390/plants12112147_

Round 1

Reviewer 1 Report

The research article under the title "Antibacterial and wound healing properties of natural products: a review of plant species with therapeutic potential against staphylococcus aureus wound infections" that is authored by Ana Elisa Belloto Morguette et al, reviews the herbal preparations that have antimicrobial and healing activities with potential for the treatment of wound infections caused by Staphylococcus aureus. The paper highlights the potential of plant extracts as sources of bioactive molecules for the discovery and development of new drugs or strategies for the treatment of wounds.  The manuscript is well written, organized, and presented in a clear and concise manner. The authors have provided a comprehensive review of the relevant literature. Based on my assessment, I strongly recommend its acceptance for publication in Plants journal in its current form.

Author Response

Reviewer 1

Comments and Suggestions for Authors

The research article under the title "Antibacterial and wound healing properties of natural products: a review of plant species with therapeutic potential against staphylococcus aureus wound infections" that is authored by Ana Elisa Belloto Morguette et al, reviews the herbal preparations that have antimicrobial and healing activities with potential for the treatment of wound infections caused by Staphylococcus aureus. The paper highlights the potential of plant extracts as sources of bioactive molecules for the discovery and development of new drugs or strategies for the treatment of wounds. The manuscript is well written, organized, and presented in a clear and concise manner. The authors have provided a comprehensive review of the relevant literature. Based on my assessment, I strongly recommend its acceptance for publication in Plants journal in its current form.

Response: Dear reviewer, thank you for taking the time to review our manuscript. We appreciate your detailed reading, constructive comments and favorable opinion of our manuscript.

Kind regards,

Sueli Fumie Yamada-Ogatta

on behalf of the authors

Reviewer 2 Report

This review article summarized the antibacterial activities of different plant species against growth of S. aureus. Some major points are raised shown below:

1.     Line 88, In a review article, “Results and Discussion” is not appropriate. Please see an example here (https://www.mdpi.com/2223-7747/12/9/1860).

2.     Table 1 only mentioned the model used in previous studies (e.g. in vitro or animal models). However, from Table 2, it is hard to understand the mechanisms of antibacterial activities of different plant species. Therefore, the major findings of these previous studies should be provided to understand the antibacterial activities of different plant species.

3.     In the 2.2 section, the author should discuss the potential mechanisms of antibacterial activities of different plant species in more details based on the results from Table 1.

4.     In Table 2 the pharmacological activities of natural products are too lengthy, please just list the important findings and the major mechanisms in Table 2. The detailed information of pharmacological activities of natural products could be discussed in the section of 2.4.

Moderate editing of English language is required.

Author Response

Reviewer 2

Response: Dear reviewer, thank you for taking the time to review our manuscript. We appreciate your detailed reading of the paper and the corresponding constructive comments that allowed us to improve the manuscript. In response to your request, we promptly respond to each comment below. In addition, we highlighted the changes in red in the main text of the manuscript

Comments and Suggestions for Authors

This review article summarized the antibacterial activities of different plant species against growth of S. aureus. Some major points are raised shown below:

  1. Line 88, In a review article, “Results and Discussion” is not appropriate. Please see an example here (https://www.mdpi.com/2223-7747/12/9/1860).

Response: Dear reviewer, thank you for the comments, and sorry for the mistake. We changed the topic according to the given example. Please see the main text of the manuscript.

  1. Table 1 only mentioned the model used in previous studies (e.g. in vitro or animal models). However, from Table 2, it is hard to understand the mechanisms of antibacterial activities of different plant species. Therefore, the major findings of these previous studies should be provided to understand the antibacterial activities of different plant species.

Response: Dear reviewer, thank you for your comments and sorry that it is not clear how the tables were arranged.

We added the following information: “Plant derived natural products are characterized by a wide diversity of chemical class and structural complexity. Terpenes, anthraquinones, tannins, flavonoids, lipids and polysaccharides are some examples of chemical classes whose antibacterial activity has been reported in different species of bacteria [44,45]. Several physicochemical characteristics can influence the mechanism of antibacterial activity of these phytochemicals, including molecular mass, solubility, polar surface area and presence of different sub-stituents [44,45]. In general, these compounds can interfere in multiple bacterial targets, including cell wall synthesis, cell membrane permeability, DNA replication, protein synthesis or enzyme activity [44]. Most of the studies presented in this review reported the main phytochemicals of plant extracts (Table S1), and in general they belong to these chemical classes.

Table 1 shows the plants that have already been studied for their antibacterial potential in S. aureus isolated from wounds (in vitro) and/or in animal models of wounds infected with S. aureus (in vivo assay), covering the period from January 2015 to March 2023. Results of studies presented in Table 1 that used traditional pharmaceutical formulations incorporated with natural plant products and evaluated their antibacte-rial effects in in vivo assay were summarized in topic 4 of this review. Studies that only performed in vitro antibacterial activity assays against S. aureus and/or did not evaluate pharmaceutical formulations incorporated with natural plant products in in vivo assays were summarized in Table S1.

  1. In the 2.2 section, the author should discuss the potential mechanisms of antibacterial activities of different plant species in more details based on the results from Table 1.

Response: Dear reviewer, thank you for your comments. Most of the studies selected for the review did not evaluate the antibacterial mechanism of action, as they worked with extracts or semi-purified fractions. When probable mechanisms of action have been investigated, we include them in the text or table and are highlighted in red.

In addition, as described above, we included the following information: “Plant derived natural products are characterized by a wide diversity of chemical class and structural complexity. Terpenes, anthraquinones, tannins, flavonoids, lipids and polysaccharides are some examples of chemical classes whose antibacterial activity has been reported in different species of bacteria [44,45]. Several physicochemical characteristics can influence the mechanism of antibacterial activity of these phytochemicals, including molecular mass, solubility, polar surface area and presence of different sub-stituents [44,45]. In general, these compounds can interfere in multiple bacterial targets, including cell wall synthesis, cell membrane permeability, DNA replication, protein synthesis or enzyme activity [44]. Most of the studies presented in this review reported the main phytochemicals of plant extracts (Table S1), and in general they belong to these chemical classes.”

Please, see the main text, Table 1 and Supplementary Table S1.

  1. In Table 2 the pharmacological activities of natural products are too lengthy, please just list the important findings and the major mechanisms in Table 2. The detailed information of pharmacological activities of natural products could be discussed in the section of 2.4.

Response: Dear reviewer, thank you for your comments. We changed the Table 2 as requested, and the changes are highligthed in red in the text and in Tabble 2.

As we have modified the tables, the numbers of some references have also changed. Furthermore, two references were added to explain probable mechanisms of action of compounds present in plant extracts.

Comments on the Quality of English Language

Moderate editing of English language is required.

Response: Dear reviewer, thank you for your comments. We reviewed the English language, as requested.

Kind regards,

Sueli Fumie Yamada-Ogatta

on behalf of the authors

Round 2

Reviewer 2 Report

All questions have been answered and the manuscript has been revised appropriately. 

Minor editing of English language required